# Dark matter effective theory

**Joachim Brod**

Department of Physics, University of Cincinnati, Cincinnati, OH 45221, USA

joachim.brod@uc.edu

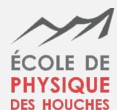

*Part of the Dark Matter*
*Session 118 of the Les Houches School, July 2021*
*published in the Les Houches Lecture Notes Series*

## Abstract

Les Houches 2021 lectures on dark matter effective field theory (short course). The aim of these two lectures is to calculate the DM-nucleus cross section for a simple example, and then generalize to the treatment of general effective interactions of spin-1/2 DM. Relativistic local operators, the heavy-DM effective theory, the chiral effective Lagrangian, and nuclear effective operators are briefly discussed.

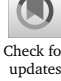

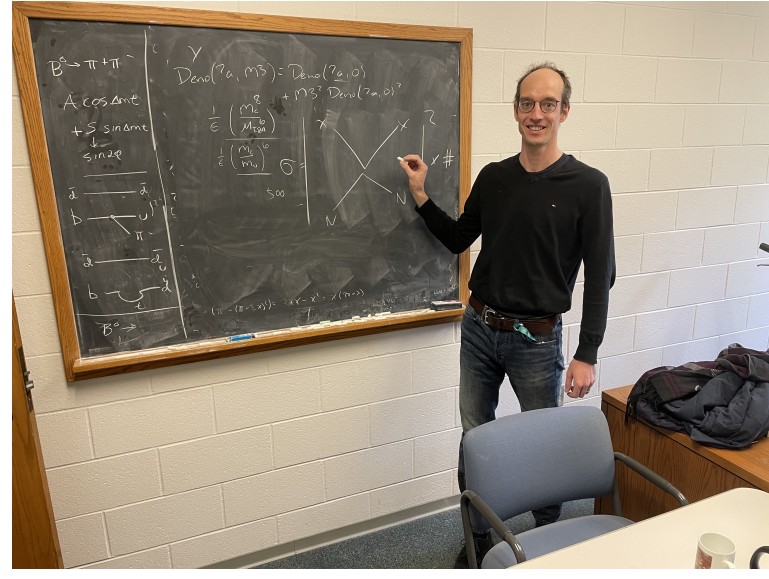

# 1 Introduction

In these two lectures on "DM effective field theories" we will mainly be concerned with calculating the differential scattering rate for DM scattering on nuclei:

$$\frac{dR}{dq} = \frac{\rho_0}{m_A m_{DM}} \int_{v_{min}} dv \, v f(v) \frac{d\sigma}{dq}(v, q). \tag{1}$$

The left side of the equation can be integrated and compared with experimental data. The right side is proportional to the local DM density $\rho_0$; in our context, we regard it as given (astrophysical input). $f(v)$ is the velocity distribution of DM in the halo; we also regard it as given. For a given nuclear target $A$ with nucleus mass $m_A$, we thus need to calculate the *differential cross section $d\sigma/dq$*.

In general, calculating the cross section is a complicated problem. It involves DM particles (whose properties like mass and spin are not known) scattering on such complex objects as atomic nuclei – strongly bound states of quarks and gluons. One way to deal with this problem is to split the calculation into many separate calculations ("factorization") that are simplified by making well-motivated approximations ("power counting"). These are essential ideas of the *effective field theory* approach.

In the first lecture, we will calculate in detail the DM scattering cross section for a vector interaction "from first principles", introducing some of the ideas of effective theory in passing. In the second lecture, we will extend these ideas into a framework that works also for more general interactions. Many advanced topics enter this discussion; unfortunately, there is no time to properly explain all these interesting concepts in these lectures. Instead, I will focus on a qualitative understanding. However, for the benefit of the interested reader, I list some introductory material (lectures, reviews, and original articles) to the various topics here. The selection is by no means complete.

Introductions to the general ideas of effective field theories are given by Pich [1] and Neubert [2]. Stewart has an online course that is publically accessible. Another general review article is the one by Georgi [3]. Buras has written an introduction to the weak effective

Hamiltonian in his Les Houches lectures [4]. The standard review by Buchalla, Buras, and Lautenbacher [5] is more concise. The lectures by Pich [6] can serve as an introduction to chiral perturbation theory. The book by Scherer and Schindler [7] has many more details and discusses also baryons. The book by Georgi [8] also contains a discussion of the chiral Lagrangian. Epelbaum discusses nuclear physics based on the chiral approach in his lecture notes [9]. The heavy-DM effective theory was mainly adapted from the heavy-quark effective theory as applied mainly in flavor physics. Manohar and Wise have written a book on the topic [10]; two further sets of lecture notes are by Neubert [11] and Buchalla [12].

## 2   Lecture 1: Calculation of a simple cross section

To set the stage, we consider the kinematics of the elastic scattering process. Let us assume we have a DM particle of mass $m_\chi = 100\,\text{GeV}$, scattering off an atomic xenon nucleus (mass roughly $m_A = 130\,\text{GeV}$). The escape velocity of our galaxy is about $500\,\text{km/s}$,[1] or in units of speed of light $500\,\text{km/s}/(299792\,\text{km/s}) \sim 0.002$. This is the maximum speed of DM in the galactic halo – DM can safely be treated as nonrelativistic. We can use energy and momentum conservation to estimate the momentum transfer in the scattering process. It will be useful for later to find combinations that are invariant under change of the coordinate system (as DM is nonrelativistic, Galilean invariance will suffice). One obvious candidate is the *momentum transfer*. Denoting the in- and outgoing DM momenta by $\boldsymbol{p}_1$, $\boldsymbol{p}_2$ and the in- and outgoing nuclear momenta by $\boldsymbol{k}_1$, $\boldsymbol{k}_2$, we define $\boldsymbol{q} \equiv \boldsymbol{k}_2 - \boldsymbol{k}_1 = \boldsymbol{p}_1 - \boldsymbol{p}_2$.

Let us calculate the maximal momentum transfer in the lab frame, scattering on a xenon nucleus at rest. We have $|\boldsymbol{p}_1| = 0.002 m_\chi = 0.2\,\text{GeV}$, $|\boldsymbol{k}_1| = 0$. The momentum transfer is maximal for a "head-on" collision, such that $\boldsymbol{p}_1$ and $\boldsymbol{p}_2$ are collinear. A straighforward calculation gives[2]

$$|\boldsymbol{q}|_{\text{max}} = \frac{2\mu_A}{m_\chi}|\boldsymbol{p}_1|, \tag{5}$$

where $\mu_A = m_A m_\chi/(m_A + m_\chi)$ is the nucleus-DM reduced mass. In our numerical example, $|\boldsymbol{q}|_{\text{max}} \sim 225\,\text{MeV}$. The maximal energy transferred to the nucleus is

$$E_{A,\text{out,max}} = \frac{4 m_A m_\chi}{(m_A + m_\chi)^2} E_{\chi,\text{in}}. \tag{6}$$

Using $E_{\chi,\text{in}} = |\boldsymbol{p}_1|^2/(2m_\chi)$, this gives $E_{A,\text{out,max}} \sim 200\,\text{keV}$.

We see that energy and momentum transfer are very small compared to the other scales in the problem, namely, the DM and nuclear masses. We can use this to make some approximations that will simplify the calculation of the cross section.

---

[1]Here, we neglect the motion of the earth; see Ref. [13] for a detailed discussion.

[2]For collinear momenta, momentum conservation gives

$$p_1 = k_2 \mp p_2 \quad \Rightarrow \quad p_2^2 = (p_1 - k_2)^2 \tag{2}$$

(the sign is negative if $m_\chi < m_A$, and positive otherwise). Energy conservation gives

$$\frac{p_2^2}{2m_\chi} + \frac{k_2^2}{2m_A} = \frac{p_1^2}{2m_\chi} \Rightarrow p_2^2 = p_1^2 - \frac{m_\chi}{m_A}k_2^2. \tag{3}$$

Substituting Eq. (3) into Eq. (2) and rearranging gives

$$2p_1 = \left(1 + \frac{m_\chi}{m_A}\right)k_2, \tag{4}$$

and solving for $k_2$ yields Eq. (5).

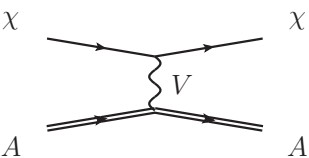

Figure 1: Leading order Feynman diagram for DM scattering on a nucleus via the exchange of a vector particle.

We did not yet talk about the force between DM and the nucleus. While the scattering process itself is nonrelativistic, any realistic, fundamental model of DM must be Lorentz invariant. In relativistic quantum field theory, forces between particles are described by exchanges of bosons. In the following, we consider a simple toy model that describes fermionic DM interactions with SM quarks via the exchange of a heavy vector particle ("a $Z'$ model"). The relevant interaction Lagrangian[3] is

$$\mathcal{L} = g_V V_\mu \bar{\chi} \gamma^\mu \chi + g'_V V_\mu \sum_{q=u,d} \bar{q} \gamma^\mu q. \tag{7}$$

Here, $u$ and $d$ denote the up- and down-quark fields, while $V^\mu$ denotes the massive vector field. For simplicity, we assume equal couplings $g'_V$ to up and down quarks, and that all other couplings to SM particles vanish. The coupling between the vector particle and DM is denoted by $g_V$. Using the Feynman rules corresponding to this Lagrangian, we can calculate the transition amplitude $\mathcal{M}$ and then obtain the differential cross section for elastic scattering, see Eq. (69) below. It will be instructive to do this calculation explicitly for our simple example.

At leading order, the transition amplitude $\mathcal{M}$ is given in terms of the S-matrix element by

$$
\begin{aligned}
S &= (2\pi)^4 i \mathcal{M} \delta^4(p_{\text{out}} - p_{\text{in}}) \\
&= \frac{(-i)^2}{2!} (-ig_V)(-ig'_V) \int d^4x d^4y \langle \chi(\boldsymbol{p}_2, r') A(\boldsymbol{k}_2, s')| \\
&\quad \times T\{(\bar{\chi} \slashed{V} \chi)(x) \sum_{q=u,d} (\bar{q} \slashed{V} q)(y) + (x \leftrightarrow y)\} |\chi(\boldsymbol{p}_1, r) A(\boldsymbol{k}_1, s)\rangle.
\end{aligned}
\tag{8}
$$

The leading-order Feynman diagram for the scattering amplitude is shown in Fig. 1.

The DM current can be contracted with the external states as usual. The vector fields can only be contracted internally and hence yield a propagator factor

$$D(x - y) = \langle 0|T\{V(x)V(y)\}|0\rangle = -i \int \frac{d^4k}{(2\pi)^4} \frac{g^{\mu\nu} - \frac{k^\mu k^\nu}{M_V^2}}{k^2 - M_V^2} e^{-ik(x-y)}. \tag{9}$$

It is relatively straightforward to see that the "gauge dependent" piece proportional to $k^\mu k^\nu$ will not contribute to the amplitude (at tree level, this follows essentially directly from translational invariance), and we will drop this piece from now on. Exchanging $x \leftrightarrow y$ in Eq. (8) cancels the factor 1/2!. Inserting the propagator gives

$$
\begin{aligned}
S &= -ig_V g'_V \int \frac{d^4k}{(2\pi)^4} \int d^4x d^4y \frac{g^{\mu\nu}}{k^2 - M_V^2} e^{-ik(x-y)} \\
&\quad \times \langle \chi(\boldsymbol{p}_2, r') A(\boldsymbol{k}_2, s')|(\bar{\chi} \gamma_\mu \chi)(x) \sum_{q=u,d} (\bar{q} \gamma_\nu q)(y)|\chi(\boldsymbol{p}_1, r) A(\boldsymbol{k}_1, s)\rangle.
\end{aligned}
\tag{10}
$$

---

[3]Strictly speaking, this is a Lagrangian density, such that the Lagrangian is given by $L = \int dx^4 \mathcal{L}$. To avoid clumsy language, I will follow common habit and do not distinguish the two where no confusion can arise.

Now we need to consider the contractions of the external states with the field operators. Since DM is completely neutral, the matrix element in the second line factorizes into two independent parts:

$$\langle \chi(\boldsymbol{p}_2, r') A(\boldsymbol{k}_2, s') | (\bar{\chi} \gamma_\mu \chi)(x) \sum_{q=u,d} (\bar{q} \gamma_\nu q)(y) | \chi(\boldsymbol{p}_1, r) A(\boldsymbol{k}_1, s) \rangle$$
$$= \langle \chi(\boldsymbol{p}_2, r') | (\bar{\chi} \gamma_\mu \chi)(x) | \chi(\boldsymbol{p}_1, r) \rangle \langle A(\boldsymbol{k}_2, s') | \sum_{q=u,d} (\bar{q} \gamma_\nu q)(y) | A(\boldsymbol{k}_1, s) \rangle . \tag{11}$$

The "DM factor" is easy to evaluate, as the Lagrangian is written directly in terms of DM fields. Application of the usual Feynman rules (in this lecture, we follow the conventions of Peskin & Schroeder [14]) gives

$$\langle \chi(\boldsymbol{p}_2, r') | (\bar{\chi} \gamma^\mu \chi)(x) | \chi(\boldsymbol{p}_1, r) \rangle = \bar{u}(\boldsymbol{p}_2, r') \gamma^\mu u(\boldsymbol{p}_1, r) e^{i(p_2 - p_1) \cdot x} . \tag{12}$$

We will now take the nonrelativistic (NR) limit. Inserting the leading expansion of the spinor function[4] in the NR limit, we find

$$u(\boldsymbol{p}, r) = \sqrt{m_\chi} \begin{pmatrix} \xi \\ \xi \end{pmatrix}_r . \tag{14}$$

Here, $\xi$ is a NR two-component spinor for DM, with $(1,0)^T$ $((0,1)^T)$ denoting spin up (down) along the $z$ axis. Recalling the definition

$$\gamma^\mu = \begin{pmatrix} 0 & \sigma^\mu \\ \bar{\sigma}^\mu & 0 \end{pmatrix} , \tag{15}$$

where $\sigma^\mu = (1, \sigma^i)$, $\bar{\sigma}^\mu = (1, -\sigma^i)$ in terms of the usual Pauli matrices, we find

$$\bar{u}(\boldsymbol{p}_2, r') \gamma^0 u(\boldsymbol{p}_1, r) = 2 m_\chi \delta_{r'r} , \tag{16}$$
$$\bar{u}(\boldsymbol{p}_2, r') \gamma^i u(\boldsymbol{p}_1, r) = 0 . \tag{17}$$

Hence, the DM factor in the limit of small momentum transfer is just

$$\langle \chi_{\text{out}}(\boldsymbol{p}_2, r') | (\bar{\chi} \gamma^\mu \chi)(x) | \chi_{\text{in}}(\boldsymbol{p}_1, r) \rangle = 2 m_\chi \delta^\mu_0 \delta_{r'r} e^{i(p_2 - p_1) \cdot x} . \tag{18}$$

The hadronic current $\langle A(\boldsymbol{q}_2, s') | (\bar{q} \gamma_\mu q)(y) | A(\boldsymbol{q}_1, s) \rangle$ requires a bit more work. The metric tensor in the propagator together with the Kronecker delta in Eq. (18) ensures that only the $\mu = 0$ component will contribute. There are two ways to proceed: One can show from the explicit expressions of the quark fields that $\bar{q} \gamma^0 q \propto a^\dagger a - b^\dagger b$ which just counts the number of quarks minus the number of antiquarks. However, this calculation is somewhat tedious. We can avoid it by following a different route that is slightly more abstract but can be generalized later: we will use a symmetry. We recognize the integral over the zero component of the quark bilinear as the conserved Noether charge, $Q_B$, of the baryon current:

$$Q_B \equiv \int d^3 \boldsymbol{y} \sum_{q=u,d} \bar{q} \gamma^0 q(y) . \tag{19}$$

This just counts the number of baryons in the initial and final states. (This is underlying the "coherent enhancement" of spin-independent scattering.) In fact, the symmetry we used in this case is exact, so we get an exact result. In general, we will use approximate symmetries, which give approximate results.

---

[4]We use the chiral representation and follow the conventions in Ref. [14]. The full solution for the Dirac spinor in these conventions is

$$u(\boldsymbol{p}) = \begin{pmatrix} \sqrt{p \cdot \sigma} \xi \\ \sqrt{p \cdot \bar{\sigma}} \xi \end{pmatrix} . \tag{13}$$

**Exercise 1** *Derive the Noether current for baryon number conservation. What is the corresponding symmetry of the SM Lagrangian?*

We see that the hadronic part of the matrix elements just counts the number of baryons minus the number of antibaryons in the nucleus (recall that quarks have baryon number $+1/3$, while antiquarks have baryon number $-1/3$). Using the results in App. A, the "nucleus factor" gives

$$
\begin{aligned}
\langle A(\boldsymbol{k}_2, s')| &= \sum_{q=u,d} \left(\bar{q}\gamma^0 q\right)(y)|A(\boldsymbol{k}_1, s)\rangle \\
&= \langle A(\boldsymbol{k}_2, s')| \sum_{q=u,d} \left(\bar{q}\gamma^0 q\right)(0) e^{i(k_2-k_1)\cdot y}|A(\boldsymbol{k}_1, s)\rangle \\
&= e^{i(k_2-k_1)\cdot y} 2k_1^0 A \delta_{s's},
\end{aligned}
\tag{20}
$$

where $A$ is the atomic number (number of nucleons) in the nucleus. Note that in the NR limit, $k_1^0 = m_A$. Now we can combine our results and find

$$
\begin{aligned}
S = &-4Ai\, g_V g'_V m_\chi m_A \delta_{r'r}\delta_{s's} \\
&\times \int \frac{d^4k}{(2\pi)^4} \int d^4x\, d^4y\, \frac{1}{k^2 - M_V^2} e^{-ik(x-y)} e^{i(p_2-p_1)\cdot x} e^{i(k_2-k_1)\cdot y}.
\end{aligned}
\tag{21}
$$

We can now easily perform the $x$ and $y$ integrals; they just yield delta functions:

$$
\begin{aligned}
S = &-4Ai\, g_V g'_V m_\chi m_A \delta_{r'r}\delta_{s's} \\
&\times \int \frac{d^4k}{(2\pi)^4} (2\pi)^4 \delta^4(p_2-p_1-k)(2\pi)^4\delta^4(k+k_2-k_1)\frac{1}{k^2-M_V^2}.
\end{aligned}
\tag{22}
$$

The momentum integration is then also easy, and we obtain

$$
S = -4Ai\, g_V g'_V m_\chi m_A \delta_{r'r}\delta_{s's}(2\pi)^4\delta^4(p_2+k_2-p_1-k_1)\frac{1}{q^2-M_V^2}.
\tag{23}
$$

The left-over delta function ensures four-momentum conservation in the usual way. We have defined $q = k_2 - k_1$ as above. We can now expand in small $q$,

$$
\frac{1}{q^2 - M_V^2} = -\frac{1}{M_V^2} \times \frac{1}{1 - q^2/M_V^2} = -\frac{1}{M_V^2}\left[1 + \mathcal{O}\!\left(\frac{q^2}{M_V^2}\right)\right],
\tag{24}
$$

and retain only the leading term; this gives

$$
S = 4Ai\frac{g_V g'_V}{M_V^2} m_\chi m_A \delta_{r'r}\delta_{s's}(2\pi)^4\delta^4(p_2+k_2-p_1-k_1).
\tag{25}
$$

By definition, the transition matrix element is $S = (2\pi)^4 i\delta^4(p_2+k_2-p_1-k_1)\mathcal{M}$, so

$$
\mathcal{M} = 4A g_V g'_V \frac{m_\chi m_A}{M_V^2}\delta_{r'r}\delta_{s's}.
\tag{26}
$$

To calculate the cross section, we need $|\mathcal{M}|^2$, average over initial spins, and sum over final spins (the spin components are not measured in direct detection experiments). The average gives a factor $1/2 \times 1/2 = 1/4$, and the sum over spins is easily performed. We find

$$
\frac{1}{4}\sum_{rr'ss'}|\mathcal{M}|^2 = 16A^2\left(g_V g'_V\right)^2\frac{m_\chi^2 m_A^2}{M_V^4},
\tag{27}
$$

and the cross section is

$$\sigma = \frac{1}{\pi}\mu_{\chi A}^2 A^2 \frac{g_V^2 g_V'^2}{M_V^4}, \tag{28}$$

where we used $s = (E_{A,\text{in}} + E_{\chi,\text{in}})^2 = (m_A + m_\chi)^2$ and introduce the reduced mass of the DM-nucleus system, $\mu_{\chi A} = m_\chi m_A/(m_A + m_\chi)$.

This was straightforward[5] but tedious! In the next lecture, we will generalize the ideas used here and introduce appropriate *effective field theories*. The main idea is to perform the simplifications before we start calculating! In this way, we can treat also harder examples.

## 3 Lecture 2: Effective field theory for DM

Now we will generalize the ideas of our toy model calculation into a more general strategy, performing the simplifications from the start. We need to generalize three different techniques that played a role in the calculation above:

1. Expansion of mediator propagator

2. NR limit of DM currents

3. Hadronic matrix elements of quark currents

We will discuss these steps in turn.

**Local DM Interactions**

The underlying idea of using effective field theory is to recognize that typically several different *energy scales* contribute to a given process. An energy scale is essentially any quantity with a mass dimension. In our example above, the scales were the particle masses $M_V$, $m_\chi$ and $m_A$, as well as the momentum transfer $q$. We can then simplify the calculation by expanding in small dimensionless ratios of these scales, and retaining only the leading terms. For instance, above we expanded the propagator and kept the leading term in $q^2/M_V^2$. The result was a contribution to the amplitude that is the same as the one obtained from the *local operator*

$$Q_{1,q}^{(6)} = (\bar{\chi}\gamma^\mu\chi)(\bar{q}\gamma_\mu q), \tag{29}$$

with a coefficient $-g_V g_V'/M_V^2$. It is easy to imagine that different types of mediators (vector, scalar, pseudoscalar, ... ) would yield different types of local interaction when expanded to leading order in momentum transfer. Instead of "integrating out" all kinds of different mediators explicitly, one can just write down all possible local interactions. The "rules of the game" are to preserve Lorentz invariance and conservation laws (such as of electric charge); they manifest themselves as *symmetries*. Writing all such operators that are linearly independent gives a *basis of operators*; the systematics is nicely explained in Ref. [15]. While there are infinitely many such operators, there is only a finite number with a given *mass dimension* (see below). Instead of writing down the full basis, we consider a second example,

$$Q_{8,q}^{(7)} = m_q(\bar{\chi}i\gamma_5\chi)(\bar{q}i\gamma_5 q). \tag{30}$$

(A full basis up to mass dimension seven can be found, e.g., in Refs. [16,17] whose numbering scheme we followed here.) The number in the superscript denotes the mass dimension of

---

[5]I should mention that we oversimplified the calculation a little bit. For large momentum transfer, the DM will be able to partially resolve the point nucleus, so we should take a "form factor" into account. See below.

the operator. Since the Lagrangian density must have mass dimension four (such that the Lagrangian is a dimensionless number), it follows that the coefficient of any dimension-six operator must be suppressed by an inverse square of a heavy mass (the vector mediator mass in our example above); generally, we write this as $1/\Lambda^2$. The dimension-seven operator is suppressed by $1/\Lambda^3$. However, this is partially just a convention. We included in the definition a factor of $m_q$ that arises in many models with scalar or pseudoscalar exchange. So we could think of this as $(m_q/\Lambda) \times (1/\Lambda^2)$, where the first factor is related to electroweak symmetry breaking, and the second factor arises from integrating out the mediator.

In general, each of these operators comes with a "Wilson" coefficient whose value depends on the UV theory. We write the Lagrangian density as

$$\mathcal{L}_\chi = \sum_{a,d} \hat{C}_a^{(d)} Q_a^{(d)}, \qquad \text{where} \quad \hat{C}_a^{(d)} = \frac{C_a^{(d)}}{\Lambda^{d-4}}, \tag{31}$$

summing over mass dimension $d$ and operators $a$. Our first example above corresponds to $C_{1,q}^{(6)} = -g_V g_V'$ and $\Lambda = M_V$, with all other Wilson coefficients zero.

**Heavy DM Effective Theory**

If DM is nonrelativistic, its energy is dominated by its mass, and it is useful to perform an expansion in powers of momentum divided by mass. We treated a simple example in our explicit calculation above. This example can be generalized as follows. Recall that the (free) DM field $\chi(x)$ satisfies the Dirac equation,

$$(i\slashed{\partial} - m_\chi)\chi(x) = 0. \tag{32}$$

This can be interpreted as follows: via Fourier transformation, $\partial_\mu$ corresponds to a four-momentum. The Dirac equation implies that the energy contains the "large mass". In the NR regime we are only interested in the kinetic energy part, since the mass does not change. It is thus convenient to get rid of the implicit large mass pieces in all energies. This can be done by splitting a general DM four-momentum into $p^\mu = m_\chi v^\mu + k^\mu$, where $v^\mu$ is the four-velocity, and the components of $k^\mu$ are small compared to $m_\chi v^\mu$. We then split the DM field correspondingly:

$$\chi(x) = e^{-im_\chi v \cdot x}\big(\chi_v(x) + X_v(x)\big), \tag{33}$$

where

$$\chi_v(x) = e^{im_\chi v \cdot x}\frac{1+\slashed{v}}{2}\chi(x), \qquad X_v(x) = e^{im_\chi v \cdot x}\frac{1-\slashed{v}}{2}\chi(x), \tag{34}$$

and we rescaled all fields by a factor $e^{im_\chi v \cdot x}$. The projectors $P_v^\pm = (1 \pm \slashed{v})/2$ generalize the usual decomposition of a spinor into "large" and "small" components in a covariant way. Essentially, they project on the particles as opposed to antiparticles, as the latter cannot be produced with nonrelativistic energies. (In nonrelativistic QM this is known as a "Foldy-Wouthuysen transformation".) Due to the rescaling, derivatives correspond to small energies and momenta. The field $\chi_v(x)$ describes the NR DM modes, while $X_v(x)$ describes the anti-particles modes.

We will "integrate them out" as follows. Multiplying Eq. (32) by $(1-\not{v})/2$ yields

$$
\frac{1-\not{v}}{2}(i\not{\partial}-m_\chi)\chi(x)=0
$$

$$
\Leftrightarrow \quad \left(i\not{\partial}\frac{1+\not{v}}{2}-iv\cdot\partial-m_\chi\frac{1-\not{v}}{2}\right)e^{-im_\chi v\cdot x}\big(\chi_v(x)+X_v(x)\big)=0
$$

$$
\Leftrightarrow \quad e^{-im_\chi v\cdot x}\left((i\not{\partial}+m_\chi\not{v})\frac{1+\not{v}}{2}-(iv\cdot\partial+m_\chi)-m_\chi\frac{1-\not{v}}{2}\right)\big(\chi_v(x)+X_v(x)\big)=0 \tag{35}
$$

$$
\Leftrightarrow \quad (i\not{\partial}+m_\chi)\chi_v(x)-(iv\cdot\partial+m_\chi)\big(\chi_v(x)+X_v(x)\big)=m_\chi X_v(x)
$$

$$
\Leftrightarrow \quad (i\not{\partial}-iv\cdot\partial)\chi_v(x)-(iv\cdot\partial+2m_\chi)X_v(x)=0\,,
$$

and thus

$$
(iv\cdot\partial+2m_\chi)X_v(x)=i\not{\partial}_\perp\chi_v(x)\,. \tag{36}
$$

We used $(1+\not{v})X_v=(1-\not{v})\chi_v=0$ and $\not{v}\not{\partial}=-\not{\partial}\not{v}+2v\cdot\partial$, and have defined $\partial_\perp^\mu\equiv\partial^\mu-v^\mu v\cdot\partial$.

Now we act with the inverse differential operators $(iv\cdot\partial+2m_\chi)^{-1}$ on Eq. (36) and obtain

$$
X_v(x)=\frac{i\not{\partial}_\perp}{iv\cdot\partial+2m_\chi}\chi_v(x)\,, \tag{37}
$$

or, inserting into Eq. (33),

$$
\chi(x)=e^{-im_\chi v\cdot x}\left[1+\frac{i\not{\partial}_\perp}{iv\cdot\partial+2m_\chi}\right]\chi_v(x)\,. \tag{38}
$$

We find the *Heavy Dark Matter Effective Theory (HDMET)* Lagrangian by replacing the fields in the relativistic Lagrangian using Eq. (38) and expanding the denominator in a power series,

$$
\frac{1}{iv\cdot\partial+2m_\chi}=\frac{1}{2m_\chi}\left[1-\frac{iv\cdot\partial}{2m_\chi}+\dots\right]\,. \tag{39}
$$

We will only write down the leading term of the Lagrangian:

$$
\mathcal{L}_{\text{HDMET}}=\bar\chi_v(iv\cdot\partial)\chi_v+\dots+\mathcal{L}_{\chi_v}\,. \tag{40}
$$

**Exercise 2** *Derive the leading term, starting from the relativistic Lagrangian*

$$
\mathcal{L}=\bar\chi(i\not{\partial}-m_\chi)\chi\,.
$$

The term $\mathcal{L}_{\chi_v}$ contains the higher dimension interaction operators. We will consider the two examples from above: the vector current and the pseudoscalar current. Other currents can be treated in the same way. We will derive only the leading terms in the expansion. For the vector current we have

$$
\bar\chi\gamma^\mu\chi=\bar\chi_v e^{im_\chi v\cdot x}\gamma^\mu e^{-im_\chi v\cdot x}\chi_v=\bar\chi_v v^\mu\chi_v+\dots\,, \tag{41}
$$

where we inserted Eq. (38) and used

$$
\frac{1+\not{v}^\dagger}{2}\gamma^0\gamma^\mu\frac{1+\not{v}}{2}=\gamma^0\frac{1+\not{v}}{2}\left[\frac{1-\not{v}}{2}\gamma^\mu+v^\mu\right]=\gamma^0\frac{1+\not{v}}{2}v^\mu\,, \tag{42}
$$

and the ellipsis denotes higher-order terms. In the lab system, $v^\mu = (1,0,0,0)$ and we recover our previous result of a contact interaction. For the pseudoscalar current, the momentum-independent term vanishes,

$$\frac{1 + \slashed{v}^\dagger}{2}\gamma^0\gamma_5\frac{1 + \slashed{v}}{2} = \gamma^0\frac{1 + \slashed{v}}{2}\frac{1 - \slashed{v}}{2}\gamma_5 = 0, \tag{43}$$

so we need to go one order higher:

$$
\begin{aligned}
\bar{\chi}i\gamma_5\chi &= \bar{\chi}_v\Big[1 - \frac{i\overleftarrow{\slashed{\partial}}_\perp}{2m_\chi}\Big]e^{im_\chi v\cdot x}i\gamma_5 e^{-im_\chi v\cdot x}\Big[1 + \frac{i\slashed{\partial}_\perp}{2m_\chi}\Big]\chi_v + \dots \\
&= \bar{\chi}_v\Big[1 - \frac{i\overleftarrow{\slashed{\partial}}_\perp}{2m_\chi}\Big]i\gamma_5\Big[1 + \frac{i\slashed{\partial}_\perp}{2m_\chi}\Big]\chi_v + \dots \\
&= \frac{\partial_\mu}{m_\chi}\Big[\bar{\chi}_v\frac{\gamma_\perp^\mu\gamma_5}{2}\chi_v\Big] + \dots = \frac{\partial_\mu}{m_\chi}\big(\bar{\chi}_v S_\chi^\mu\chi_v\big) + \dots,
\end{aligned}
\tag{44}
$$

with $\gamma_\perp^\mu = \gamma^\mu - v^\mu v\cdot\gamma$. We see that this interaction is "momentum suppressed" in the low-energy limit.

**Exercise 3** *Show the validity of the last equality in the heavy-DM limit. The relativistic generalization of the spin operator is defined as $S^\mu \equiv -\frac{1}{2}\epsilon^{\mu\nu\rho\lambda}J_{\nu\rho}v_\lambda$, with the sign convention $\epsilon^{0123} = +1$ for the Levi-Civita tensor, and $J^{\mu\nu} = \frac{1}{2}\sigma^{\mu\nu}$ with $\sigma^{\mu\nu} = \frac{i}{2}[\gamma^\mu,\gamma^\nu]$ for spin 1/2. (This exercise is somewhat tedious.)*

## Chiral Effective Theory

Passing to the NR limit was straightforward for the DM currents (as long as we consider DM to be elementary). On the other hand, it is not very useful to go to the NR limit for the quark fields, since QCD is strongly coupled at low energies.

To obtain a physical amplitude, we really need to calculate a matrix element between external nucleus states. The first step is to look at the scattering on a single nucleon at a time (a justification will be given later). We can express these single-nucleon matrix elements exploiting all available symmetries in terms of so-called *form factors*. For instance, the general (elastic) vector-vector interaction can be parameterized as

$$\langle N'|\bar{q}\gamma^\mu q|N\rangle = \bar{u}_N'\Big[F_1^{q/N}(q^2)\gamma^\mu + \frac{i}{2m_N}F_2^{q/N}(q^2)\sigma^{\mu\nu}q_\nu\Big]u_N, \tag{45}$$

where $F_{1,2}^{q/N}(q^2)$ are the form factors – functions of the four-momentum transfer $q^2$. We have seen above that, e.g., $F_1^{u/p}(0) = 2$ and $F_1^{d/p}(0) = 1$.

Similarly form factors can be written down for other interactions ($V-A$, scalar, ...). However, the determination of the functions $F(q^2)$ is not always as simple. For the electromagnetic vector current, they can be measured in processes with photon exchange (e.g. elastic electron-nucleon scattering). Some others can by measured in neutrino scattering. Some, however, cannot (currently) be measured.

A systematic approach is to exploit the *chiral symmetry of QCD*. I will give only the very basic idea of what is needed; see, for instance, Refs. [6–8] for details. The QCD Lagrangian for three massless quarks $q = (u,d,s)$ is

$$\mathcal{L}_{\text{light quark}} = \bar{q}i\slashed{D}q = \bar{q}_L i\slashed{D}q_L + \bar{q}_R i\slashed{D}q_R, \tag{46}$$

where $D_\mu = \partial_\mu + i g_s T^a G_\mu^a$ is the covariant derivative of QCD. This Lagrangian is invariant under *chiral rotations* of the quark fields

$$q_L \to L\, q_L\,, \qquad q_R \to R\, q_R\,, \tag{47}$$

where $L \in SU(3)_L, R \in SU(3)_R$.

**Exercise 4** *Verify this explicitly. Show that quark mass terms break this symmetry.*

At low energies, QCD is strongly coupled and the dynamics is (so far) not analytically understood. However, we know that the chiral symmetry $SU(3)_L \times SU(3)_R$ is spontaneously broken (in addition to the explicit breaking by quark masses and QED effects). This is enough to write down the most general effective theory of QCD in terms of its low-energy degrees of freedom, the pions and nucleons. We collect the pions into the matrix $U = \exp\left(i\sqrt{2}\Pi/f\right)$, where

$$\Pi = \begin{pmatrix} \frac{\pi^0}{\sqrt{2}} + \frac{\eta_8}{\sqrt{6}} & \pi^+ & K^+ \\ \pi^- & -\frac{\pi^0}{\sqrt{2}} + \frac{\eta_8}{\sqrt{6}} & K^0 \\ K^- & \bar{K}^0 & -\frac{2\eta_8}{\sqrt{6}} \end{pmatrix} \tag{48}$$

contains the Goldstone-boson fields, and $f = f_\pi \simeq 92\,\text{MeV}$ can be identified with the pion decay constant. This matrix is unitary and transforms as $U \to R U L^\dagger$ under $SU(3)_L \times SU(3)_R$. We use it to construct an effective Lagrangian for the pion fields that is invariant under these rotations. This Lagrangian will be non-renormalizable, and the pion fields will transform nonlinearly under axial rotations, reflecting the spontaneous breaking of the symmetry. The leading-order (LO) term is simply

$$\mathcal{L}_{\text{ChPT,LO}} = \frac{f^2}{4} \text{Tr}\left(\partial_\mu U^\dagger \partial^\mu U\right). \tag{49}$$

Apart from a constant, there is no chirally invariant term without derivatives. Effectively, the chiral Lagrangian is an expansion in derivatives, or, in Fourier space and using our estimate above, in small momenta. Chiral symmetry ensures that all terms are proportional to the same factor $f^2$.

Explicit breaking effects can also be included. Consider the quark masses. The mass term

$$\mathcal{L}_{\text{quark mass}} = -\bar{q} M_q q = -\bar{q}_L M_q q_R + \text{h.c.}\,, \tag{50}$$

with $M_q = \text{diag}(m_u, m_d, m_s)$ is formally invariant under chiral rotations if we let the quark mass matrix transform as $M_q \to L M_q R^\dagger$. Recalling the transformation law $U \to R U L^\dagger$, we see that the corresponding mass term in the effective theory must be proportional to $\text{Tr}\left[M_q U\right] + \text{h.c.}$. Similar to Eq. (50), this term is Hermitean and formally invariant under chiral rotations. Only its coefficient cannot be predicted by chiral symmetry. Using the fact that the quark mass matrix is actually real, the chiral Lagrangian including the mass term is given by

$$\mathcal{L}_{\text{ChPT,LO}} = \frac{f^2}{4} \text{Tr}\left(\partial_\mu U^\dagger \partial^\mu U\right) + \frac{B_0 f^2}{2} \text{Tr}\left[M_q\left(U + U^\dagger\right)\right], \tag{51}$$

where and $B_0 \sim 2.67\,\text{GeV}$ is another low-energy constant that cannot be predicted from symmetry arguments. (It is related to the "quark condensate" and can be determined, e.g., by lattice QCD.) This Lagrangian can be used to show that $m_\pi^2 \propto m_q$ and hence to extract the quark mass ratios from experimental data. The procedure followed here of translating symmetry breaking terms from the partonic to the effective Lagrangian is sometimes called "spurion method".

In a similar way, DM matter interactions with pions can be written down. For instance, let us consider our previous two examples – the couplings to pions of a DM vector current, $v^\mu \bar{\chi}_v \chi_v$, coupled to a quark vector current, and a DM pseudoscalar current, $\partial_\mu(\bar{\chi}_v S^\mu_\chi \chi_v)/m_\chi$, coupled to a quark pseudoscalar current – as in Eqs. (29) and (30). Treating the DM current as spurions like we did for the quark mass terms, and keeping only leading terms, the chiral Lagrangian can be shown to be [16]

$$
\begin{aligned}
\mathcal{L}_{\chi,\text{ChPT}} \supset & -\frac{if^2}{2}\text{Tr}\Big[\big(U(v\cdot\partial)U^\dagger + U^\dagger(v\cdot\partial)U\big)\overline{C}_1^{(6)}\bar{\chi}_v\chi_v\Big] \\
& -\frac{B_0 f^2}{2}\text{Tr}\Big[(U-U^\dagger)M_q\overline{C}_8^{(7)}\frac{i\partial_\mu}{m_\chi}(\bar{\chi}_v S^\mu_\chi \chi_v)\Big],
\end{aligned}
\tag{52}
$$

where $\overline{C}_i^{(d)} = \text{diag}\big(C_{i,u}^{(d)}, C_{i,d}^{(d)}, C_{i,s}^{(d)}\big)$. Note that these terms are engineered to be Hermitian and even under parity (where $U \to U^\dagger$). Expanding the $U$ matrices in inverse powers of $f$, it is easy to see that the leading coupling of the vector current is to a pair of pions, while the pseudoscalar current couples to a single pion.

**Exercise 5** *Perform the expansion explicitly to leading order in $1/f$.*

Our treatment of low-energy QCD for DM scattering is of course not complete without the inclusion of nucleons. We will just quickly summarize the results here; see, e.g., Ref. [16] for details.

The first step is to pass to the low-energy limit also for nucleons. This approximation is valid as long as $\boldsymbol{q} \ll m_N$, with $m_N \approx 1\,\text{GeV}$ the nucleon mass. This is done by splitting the baryon momentum $p^\mu$ into $p^\mu = m_N v^\mu + k^\mu$, with $k^\mu$ the small residual momentum, and introducing a rescaled nucleon field [18]

$$
B_v(x) = \exp(im_N \slashed{v}\, v\cdot x)B(x).
\tag{53}
$$

The octet of baryon fields forms a $3 \times 3$ matrix, given by

$$
B_v = \begin{pmatrix}
\frac{1}{\sqrt{2}}\Sigma^0_v + \frac{1}{\sqrt{6}}\Lambda_v & \Sigma^+_v & p_v \\
\Sigma^-_v & -\frac{1}{\sqrt{2}}\Sigma^0_v + \frac{1}{\sqrt{6}}\Lambda_v & n_v \\
\Xi^-_v & \Xi^0_v & -\frac{2}{\sqrt{6}}\Lambda_v
\end{pmatrix}.
\tag{54}
$$

If we are interested only in tree-level processes, we can drop the states other than the proton and neutron and just use

$$
B_v = \begin{pmatrix}
0 & 0 & p_v \\
0 & 0 & n_v \\
0 & 0 & 0
\end{pmatrix}.
\tag{55}
$$

The interaction with the DM currents can be constructed using a spurion method similar to what we did for the pions. See Ref. [16] for details and general expressions.[6] Here, we will just collect the pieces that are relevant for the vector and pseudoscalar currents, discussed above:

$$
\mathcal{L}^{(0)}_{\chi,\text{HBChPT}} \supset (\bar{\chi}_v \chi_v)(\bar{p}_v p_v)\big(2\hat{C}_{1,u}^{(6,0)} + \hat{C}_{1,d}^{(6,0)}\big) + (p_v \leftrightarrow n_v, u \leftrightarrow d).
\tag{56}
$$

Notice that this is essentially the same result as we obtain using the baryon number conservation argument above – no surprises here! Due to parity conservation, there is no direct coupling

---

[6]For baryons, there is considerable freedom in choosing the transformation under $SU(3)_L \times SU(3)_R$, as long as they transform as an octet under the unbroken "vectorial" part. See Ref. [8] for a nice discussion.

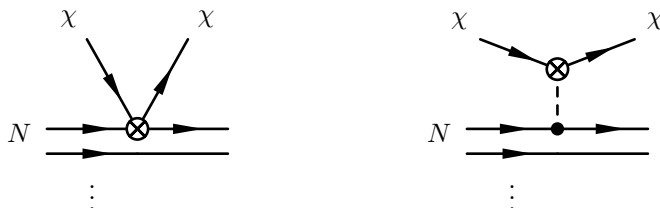

Figure 2: Leading-order diagrams for DM-nucleus scattering. The effective DM–nucleon and DM–pion interactions are denoted by a circle, the dashed lines denote mesons, and the dots represent the remaining nucleon lines.

of the pseudoscalar current to a pure single-nucleon current; rather, the leading term has the schematic form $\partial(\bar{\chi}_v S_\chi \chi_v)(\bar{p}_v p_v)\pi + \dots$. However, we have seen above that the pseudoscalar current couples to a single pion. Hence we need the pion-nucleon interaction,

$$\mathcal{L}_{\text{HBChPT}}^{(1),\text{QCD}} \supset \frac{g_A}{f} \partial_\mu\big(\bar{p}_v S_N^\mu p_v - \bar{n}_v S_N^\mu n_v\big)\pi^0 , \tag{57}$$

with the pion-nucleon coupling $g_A = 1.2756(13)$.

We now have all the ingredients to write Feynman diagrams for the scattering, see Fig. 2. How do we decide which of these diagrams are important? What about loops? Does the perturbative expansion even make sense in the strongly-coupled regime? To answer this question, recall that chiral perturbation theory is not a coupling expansion, but a derivative expansion. Any chirally invariant term without derivatives is just a constant and hence irrelevant. Now consider a general amplitude involving an arbitrary number of pions, carrying momenta at most of order $Q \ll m_N, m_\chi$. What is the overall scaling $(Q/M)^\nu$ of the amplitude, where $M$ is either $m_N$ or $m_\chi$? Let us count: each pion propagator contributes a factor $1/Q^2$. Each loop integral contributes a factor $Q^4$. Each derivative in the interaction vertices contributes a factor $Q$. So, for a diagram with $L$ loops, $V_i$ interaction vertices with $d_i$ derivatives, and $I$ internal lines, we find

$$\nu = \sum_i V_i d_i - 2I + 4L . \tag{58}$$

Now it is known since Euler that in a given (connected) diagram, the numbers of lines ($I$), vertices ($V_i$), and faces ($L$) is constrained by the relation

$$L - I + \sum_i V_i = 1 . \tag{59}$$

**Exercise 6** *Can you find a proof for this relation?*

Inserting the relation (59) into Eq. (58) yields

$$\nu = \sum_i V_i(d_i - 2) + 2L + 2 . \tag{60}$$

Since $L \geq 0$, we see that the leading diagrams are tree-level diagrams. Now this relation is strictly valid only for massless pions with only self interactions. We now need to include pion masses, and interaction with nucleons and DM. Since $Q^2$ will be at least of order $m_\pi^2$, and $m_\pi^2 \sim m_q$, we should modify the power counting relation to ($m_i$ are the factors of quark masses in vertex $V_i$)

$$\nu = \sum_i V_i(d_i + 2m_i - 2) + 2L + 2 . \tag{61}$$

Next, we include the nucleons. Since we treat them as nonrelativistic, their propagators scale differently from the pion propagators:

$$\frac{-i(\not{p}+\not{q}+m_N)}{(p+q)^2-m_N^2} \xrightarrow{q\to 0} \frac{-i(\not{p}+m_N)}{2p\cdot q}, \tag{62}$$

where we used $p^2 = m_N^2$. Hence, each nucleon propagator scales as $1/Q$. Here, $p$ is the nucleon momentum and $q$ is the additional momentum induced by pion interactions. (Note that the numerator in the second term in Eq. (62) is proportional to the projector $P_v^+$.) Hence, a general Feynman diagram with $I_\pi$ internal pion lines and $I_N$ internal nucleon lines scales as $Q^\nu$, where

$$\nu = \sum_i V_i(d_i + 2m_i) - 2I_\pi - I_N + 4L + d_\chi. \tag{63}$$

Here, I added the chiral dimension of the DM vertex (we will always consider one single DM interaction for nuclear scattering). Now we can again use the topological relation

$$L - I_\pi - I_N + \sum_i V_i = 1, \tag{64}$$

together with

$$2I_N + E_N = \sum_i V_i n_i, \tag{65}$$

where $n_i$ is the number of nucleon fields in the interaction $i$, to obtain

$$\nu = \sum_i V_i\left(d_i + 2m_i + \frac{n_i}{2} - 2\right) + 2L - \frac{E_N}{2} + 2 + d_\chi. \tag{66}$$

Finally, we note that the relations (59), (64) (65) are valid for each connected component of the graph. Moreover, since the nucleons are nonrelativistic, their number is conserved in the scattering process. So we can sum over the connected components of the graph and replace $E_N$ by two times the number of nucleon lines, $A$, and obtain the final version of the power counting formula for a $A$-nucleon irreducible graph with $C$ connected components:

$$\nu = 4 - A - 2C + 2L + \sum_i V_i\left(d_i + 2m_i + \frac{n_i}{2} - 2\right) + d_\chi. \tag{67}$$

This power counting formula works well in our case where there a single nucleon involved in the scattering. It tells us that the leading contributions to the scattering amplitude come from tree-level diagrams with single-nucleon interactions. Numerically, the suppression is of order 30% for each power of momentum [16] – taken to be somewhat larger than expected in pure ChPT in order to account for additional effects of the bound-state nucleons.

Going back to our two examples, the leading diagrams for DM with a vector mediator and pseudoscalar mediator are given in Fig. 2, left and right panel, respectively.

Finally, we need the nuclear matrix elements. The interactions are given effectively in terms of nonrelativistic nucleon fields, possibly contracted with spin matrices and derivatives. Again, one can write down all possible allowed interaction that are now assumed to be Galilean invariant. Here, we are only interested in interactions with single nucleons and we can follow Ref. [19]. The two operators we need are

$$\mathcal{O}_1^N = \mathbb{1}_\chi \mathbb{1}_N, \qquad \mathcal{O}_6^N = \left(\vec{S}_\chi \cdot \frac{\vec{q}}{m_N}\right)\left(\vec{S}_N \cdot \frac{\vec{q}}{m_N}\right). \tag{68}$$

The spin and unit matrices act on two-dimensional NR spinor space. The NR field operators are not written explicitly due to some unfortunate convention. More operators can be found

in Ref. [19], and their connection to the UV DM interactions is given in Ref. [16]. At first sight, it might seems like the second operator is suppressed by $\vec{q}^{\,2}/m_N^2$; recall, however, that the second diagram in Fig. 2 has a pion propagator that contributes a factor $1/(\vec{q}^{\,2}+m_\pi^2)$, thus largely canceling the suppression (see also the explicit result in Eq. (75) below).

Fitzpatrick et al. have determined the nuclear matrix elements for a variety of isotopes used in direct detection experiments (xenon, germanium, fluorine, iodine, sodium) using shell model calculations. The results are given numerically in Ref. [19] in terms of nuclear form factors. They can be used to calculate the NR differential cross section [20]:[7]

$$\frac{d\sigma}{dE_R} = \frac{m_A}{2\pi v^2}\frac{1}{2j_\chi+1}\frac{1}{2j_A+1}\sum_{\text{spins}}|\mathcal{M}|^2 \equiv \frac{m_A}{2\pi v^2}\sum_{ij}\sum_{N,N'=p,n}c_i^N c_j^{N'}F_{ij}^{(N,N')}. \tag{69}$$

Here, the $c_i^N$ are the coefficients of the nuclear operators $\mathcal{O}_i^N$. For our two cases, the nuclear form factors are given by $F_{1,1}^{(N,N')} = F_M^{(N,N')}$ and $F_{6,6}^{(N,N')} = (\vec{q}^{\,4}/16)F_{\Sigma''}^{(N,N')}$, in the notation of Ref. [19]. For instance, for scattering on the most abundant xenon isotope $^{132}$Xe:

$$F_M^{(p,p)} = e^{-2y}\left(2.9 - 11y + 15y^2 - 10y^3 + 4y^4 - 0.88y^5 + 0.11y^6 + \dots\right)\times 10^3\,, \tag{70}$$

while $F_{\Sigma''}^{(p,p)} = 0$ because $^{132}$Xe has spin zero. Here, $b = \sqrt{41.467/(45A^{-1/3}-25A^{-2/3})}\,\text{fm}$. The (similar) form factors $F_M^{(p,n)}$ and $F_M^{(n,n)}$, as well as the form factor for the other isotopes, can be found in Ref. [19]. For a realistic cross section, we should weight the xenon isotopes by their natural abundances.

It is instructive to also look at the scattering on fluorine, $^{19}$F:

$$F_M^{(p,p)} = e^{-2y}\left(81 - 96y + 36y^2 - 4.7y^3 + 0.19y^4\right). \tag{71}$$

Note that $2900/81 = 35.8 \sim (54/9)^2 = 36$, this is again the coherent enhancement. Fluorine has nuclear spin $1/2$ and hence

$$F_{\Sigma''}^{(p,p)} = e^{-2y}\left(0.903 - 2.37y + 2.35y^2 - 1.05y^3 + 0.175y^4\right) \tag{72}$$

is non-zero.

It should now be clear how to calculate the general scattering cross section. For our vector mediator example, we have

$$c_1^p = 2\hat{C}_{i,u}^{(6)} + \hat{C}_{i,d}^{(6)}\,, \quad c_1^n = \hat{C}_{i,u}^{(6)} + 2\hat{C}_{i,d}^{(6)}\,, \tag{73}$$

or, inserting the explicit values,

$$c_1^p = c_1^n = -3\frac{g_V g_V'}{M_V^2}\,. \tag{74}$$

Inserting this into Eq. (69) immediately gives the differential cross section. For the pseudoscalar interaction we find

$$\begin{aligned}
c_6^p &= -\frac{B_0 m_N^2}{m_\chi}\frac{g_A}{m_\pi^2+\vec{q}^{\,2}}\left(m_u\,\hat{C}_{8,u}^{(7)} - m_d\,\hat{C}_{8,d}^{(7)}\right)\,, \\
c_6^n &= \frac{B_0 m_N^2}{m_\chi}\frac{g_A}{m_\pi^2+\vec{q}^{\,2}}\left(m_u\,\hat{C}_{8,u}^{(7)} - m_d\,\hat{C}_{8,d}^{(7)}\right).
\end{aligned} \tag{75}$$

Note that the coefficients are *momentum dependent*. This has to be taken into account when integrating the differential cross section over the respective energy sensitivity windows for the different experiments.

---

[7]To switch to the cross section differential in the momentum transfer, note that $q = \sqrt{2E_R m_A}$.

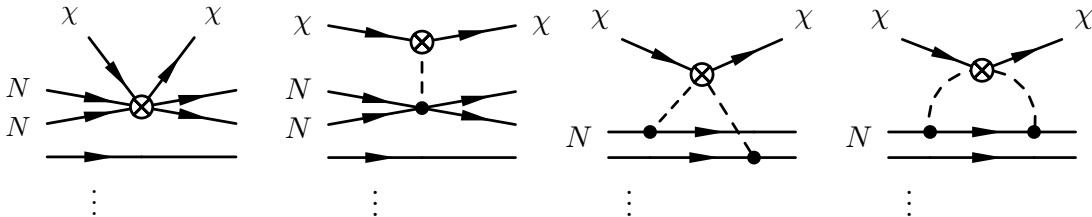

Figure 3: Sample NLO diagrams for the DM-nucleon scattering inside nuclei. The effective DM–nucleon or DM–meson interaction is denoted by a box, the dashed lines denote pions.

The whole chain of steps is quite straightforward to carry out even for more general interactions, even though it is tedious. Fortunately, public computer code is available to perform these tasks. The program `DirectDM` [21] can be used to calculate the coefficients of the nuclear operators given the UV interactions in terms of Wilson coefficients. The code can be downloaded at

https://directdm.github.io

Given the nuclear coefficients, `DMFormFactor` then allows for the automatic calculation of the nuclear cross section [20].

So far, we were concerned only with the leading approximation. Until we have identified the precise nature of DM, this should be sufficient. Nevertheless, we want to give an outlook on various corrections that can occur. Some of the most widely studied corrections are two-nucleon currents and perturbative radiative corrections.

Some Feynman diagrams with higher-order nuclear contributions are shown in Fig. 3. The local two-nucleon interactions (left two panels in Fig. 3) are always suppressed by three additional powers of momentum and are negligible. For certain interactions, diagrams with loops or two single-nucleon interaction (right two panels in Fig. 3) may be suppressed by only one power of momentum. This is the case for axialvector-vector, scalar-scalar, and pseudoscalar-scalar interactions. Two-nucleon currents can also be important if the leading contributions are absent in specific models. More details and explicit results can be found in Refs. [22, 23].

Similarly, radiative corrections can have a large impact if models are tuned such that leading contributions to nuclear scattering are absent by construction. As a very simple example, consider DM with only *leptophilic* interactions – DM couples only to leptons. At first sight, it seems that scattering on atomic nuclei would be absent, but single-photon exchange induces couplings to all fermions [24, 25] (see Fig. 4). More generally, the nuclear matrix elements exhibit large hierarchies (spin-dependent vs. spin-independent; momentum / velocity suppression). Whenever a contribution to a large matrix element is not generated at leading order in a model, but is not forbidden by a symmetry, it may be generated via radiative corrections. For instance, the electroweak and Yukawa interactions of the SM break parity, and loop-induced contributions to nuclear scattering can be larger than the leading terms by orders of magnitude (see Refs. [26–28] for an example with top quarks). The most important effects for fermionic DM are included in the `DirectDM` code [29].

## 4 Conclusion

We have seen that using effective field theory is very suitable in describing the nonrelativistic scattering process of dark matter on atomic nuclei. It allows us to calculate the event rates

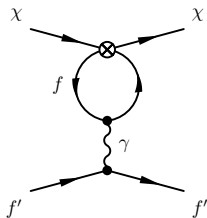

Figure 4: The mixing of dimension-six four-fermion operators into each other via the photon penguin insertion.

for a wide class of dark matter interactions in term of UV parameters and a handful of low-energy constants. Frequently, "Dark Matter Effective Field Theory" refers specifically to the formalism developed in Ref. [19]. In my opinion, this viewpoint is too restricted and has a number of drawbacks:[8] (a) Dark matter models are typically formulated in terms of elementary particle fields (quarks, leptons, gluons, . . . ), not in terms of nucleon fields. The connection between these fundamental interactions (summarized in terms of local operators and Wilson coefficients) should be made explicit. (b) The coefficients of the nuclear operators are often assumed to be constant, which is frequently not the case (see, for instance, Eq. (75)). This becomes manifest if one considers the whole tower of effective theories appropriate for dark matter direct detection. (c) Radiative corrections can have a large impact on the event rate in certain circumstances. These corrections need to be calculated in a partonic effective theory.

The practical take-home message is the following. The most convenient "meeting point" between the hadronic physics and dark-matter model building (for instance, when presenting experimental results in terms of global fits) seems to be the partonic effective theory in terms of local operators defined at a scale $\mu = 2\,\text{GeV}$, just above the chiral symmetry breaking scale of QCD. The connection to lower-scale physics can then be obtained via the chiral approach described in these lectures. The connection to realistic dark-matter models is equally straightforward, employing the usual perturbative techniques of matching and renormalization-group evolution, thus capturing the leading radiative corrections [29].

## Acknowledgements

The author would like to thank León Manuel de la Vega, as well as the anonymous referees of the lecture notes, for very valuable comments on the manuscript.

## A   Baryon number conservation

We are interested in the general form of matrix elements of the baryon current

$$\mathcal{J}_B^\mu(x) = \sum_q \bar{\psi}_q \gamma^\mu \psi_q. \tag{76}$$

Translational invariance implies

$$\langle \boldsymbol{p}', \sigma' | \mathcal{J}^\mu(x) | \boldsymbol{p}, \sigma \rangle = e^{-i(p-p')\cdot x} \langle \boldsymbol{p}', \sigma' | \mathcal{J}^\mu(0) | \boldsymbol{p}, \sigma \rangle. \tag{77}$$

(This can be derived as follows. In position space, the momentum operator acts as

$$[P_\mu, O(x)] = -i \frac{\partial}{\partial x^\mu} O(x), \tag{78}$$

---

[8]I should hasten to point out that many of these issues have already been mentioned in Ref. [19].

on any local field operator $O(x)$. It follows

$$\langle\beta|[P_\mu, O(x)]|\alpha\rangle = (p_\beta - p_\alpha)_\mu \langle\beta|O(x)|\alpha\rangle = -i\frac{\partial}{\partial x^\mu}\langle\beta|O(x)|\alpha\rangle, \tag{79}$$

and so

$$\langle\beta|O(x)|\alpha\rangle = \exp[i(p_\beta - p_\alpha)\cdot x]\langle\beta|O(0)|\alpha\rangle, \tag{80}$$

which proves Eq. (77).) Setting $\mu = 0$ in Eq. (77) and integrating over $\boldsymbol{x}$ gives

$$\langle\boldsymbol{p}', \sigma'|Q_B|\boldsymbol{p}, \sigma\rangle = (2\pi)^3\delta^3(\boldsymbol{p}' - \boldsymbol{p})\langle\boldsymbol{p}', \sigma'|\mathcal{J}^0(0)|\boldsymbol{p}, \sigma\rangle, \tag{81}$$

where we used the definition of (baryon) charge $Q_B \equiv \int d^3\boldsymbol{x}\,\mathcal{J}^0(x)$. Denoting the charge of the state $|\boldsymbol{p}, \sigma\rangle$ by $B$, we find

$$\langle\boldsymbol{p}, \sigma'|\mathcal{J}^0(0)|\boldsymbol{p}, \sigma\rangle = 2p^0 B\delta_{\sigma'\sigma}, \tag{82}$$

where $p^0 \equiv \sqrt{\boldsymbol{p}^2 + m^2}$.

# B  Solutions to exercises

### Exercise 1

The Lagrangian

$$\mathcal{L}_{\text{quark}} = \bar{q}i\slashed{D}q - \bar{q}M_q q \tag{83}$$

is invariant under the phase transformation $q \to e^{i\epsilon}q$, with $\epsilon$ real. Infinitesimally, $q \to q + i\epsilon q$. Noether's theorem tells us that for an infinitesimal symmetry $\psi \to \psi + i\epsilon\mathcal{F}$, the current

$$\mathcal{J}^\mu = -i\frac{\partial\mathcal{L}}{\partial(\partial_\mu\psi)}\mathcal{F} \tag{84}$$

is conserved. This gives

$$\mathcal{J}_B^\mu = \bar{q}\gamma^\mu q. \tag{85}$$

### Exercise 2

Inserting Eq. (38) into the Lagrangian and keeping only the leading terms gives

$$\begin{aligned}\bar{\chi}(i\slashed{\partial} - m_\chi)\chi &\to \bar{\chi}_v e^{im_\chi v\cdot x}(i\slashed{\partial} - m_\chi)e^{-im_\chi v\cdot x}\chi_v \\ &= \bar{\chi}_v(i\slashed{\partial} + m_\chi\slashed{v} - m_\chi)\chi_v = \bar{\chi}_v(iv\cdot\partial)\chi_v,\end{aligned} \tag{86}$$

where we used $\slashed{v}\chi_v = \chi_v$ in the second-to-last and Eq. (42) in the last step.

### Exercise 3

Let us first check that $S^\mu$ is indeed the spin operator. We work in the rest frame, $v = (1, 0, 0, 0)$, and using the chiral representation of the Dirac matrices:

$$\begin{aligned}S^i &= -\frac{1}{2}\epsilon^{ijk0}J_{jk} = \frac{1}{2}\epsilon^{0ijk}\frac{i}{4}[\gamma_j, \gamma_k] = \frac{i}{8}\epsilon^{0ijk}[\gamma^j, \gamma^k] = \frac{i}{8}\epsilon^{ijk}2i\epsilon^{kjl}\begin{pmatrix}\sigma^l & 0 \\ 0 & \sigma^l\end{pmatrix} \\ &= \frac{1}{2}\delta^{il}\begin{pmatrix}\sigma^l & 0 \\ 0 & \sigma^l\end{pmatrix},\end{aligned} \tag{87}$$

or

$$\vec{S} = \frac{1}{2}\begin{pmatrix} \vec{\sigma} & 0 \\ 0 & \vec{\sigma} \end{pmatrix}. \tag{88}$$

We define $\gamma_5 = i\gamma^0\gamma^1\gamma^2\gamma^3 \equiv -\frac{i}{4!}\epsilon^{\mu\nu\rho\sigma}\gamma_\mu\gamma_\nu\gamma_\rho\gamma_\sigma$. From this you can show that $\sigma^{\mu\nu}\gamma_5 = i\epsilon^{\mu\nu\rho\sigma}\sigma_{\rho\sigma}/2$. Hence we can show that

$$S^\mu = \frac{i}{2}\sigma^{\mu\sigma}\gamma_5 v_\sigma, \tag{89}$$

so that in the rest frame (using again the chiral representation)

$$\vec{S} = \frac{1}{2}\begin{pmatrix} \vec{\sigma} & 0 \\ 0 & \vec{\sigma} \end{pmatrix}, \tag{90}$$

as before. At last, we note that between NR spinors we can use the projector $P_v^+ = (1 + \not{v})/2$. We have

$$[P_v, \gamma^\mu] = \frac{1 + \not{v}}{2}\gamma^\mu - \gamma^\mu\frac{1 + \not{v}}{2} = v^\mu + \gamma^\mu\frac{1 - \not{v}}{2} - \gamma^\mu\frac{1 + \not{v}}{2} = v^\mu - \gamma^\mu\not{v}. \tag{91}$$

We can use this to show

$$P_v^+ S^\mu P_v^+ = \frac{i}{2}P_v^+ \sigma^{\mu\sigma}P_v^- \gamma_5 v_\sigma = \frac{i}{2}P_v^+\frac{i}{2}(2v^\mu v\cdot\gamma - 2\gamma^\mu)\gamma_5 P_v = \frac{1}{2}\gamma_\perp^\mu\gamma_5. \tag{92}$$

We have used $P_v^+ P_v^- = 0$ and $\not{v}P_v^- = -P_v^-$ in intermediate steps.

## Exercise 4

The light-quark Lagrangian is

$$\mathcal{L}_{\text{light quark}} = \bar{q}_L i\not{D}q_L + \bar{q}_R i\not{D}q_R. \tag{93}$$

The first term transforms as

$$\bar{q}_L i\not{D}q_L \to \bar{q}_L L^\dagger i\not{D}Lq_L = \bar{q}_L i\not{D}q_L, \tag{94}$$

and similarly for the second term. The quark mass term transforms as

$$-\bar{q}_L M_q q_R + \text{h.c.} \to -\bar{q}_L L^\dagger M_q R q_R + \text{h.c.}. \tag{95}$$

Note that the quark mass term is invariant for $L = R$ if all quark masses are equal.

## Exercise 5

We have $U = \exp\left(i\sqrt{2}\Pi/f\right)$, so expanding the exponential

$$U = 1 + i\frac{\sqrt{2}\Pi}{f} + \dots, \tag{96}$$

we see that

$$\begin{aligned}
&U(v\cdot\partial)U^\dagger + U^\dagger(v\cdot\partial)U \\
&= \left(1 + i\frac{\sqrt{2}\Pi}{f}\right)\left(-i\frac{\sqrt{2}(v\cdot\partial)\Pi}{f}\right) + \left(1 - i\frac{\sqrt{2}\Pi}{f}\right)\left(i\frac{\sqrt{2}(v\cdot\partial)\Pi}{f}\right) + \dots \\
&= 2\frac{\Pi(v\cdot\partial)\Pi}{f^2} + \dots,
\end{aligned} \tag{97}$$

quadratic in pion fields, and

$$U - U^\dagger = i\frac{2\sqrt{2}\Pi}{f} + \dots, \tag{98}$$

linear in pion fields.

**Exercise 6**

Try mathematical induction.

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
