# Peer review of "Dark Matter Effective Theory"

_SciPost Physics Lecture Notes, doi:SciPost Phys. Lect. Notes 38 (2022)_

## Round 1 · Referee Report · Anonymous · 2021-9-27

Strengths

1) clarity

2) pedagogical introduction to the subject

3) phenomenological relevance

Weaknesses

None

Report

This work collects lecture notes from the Les Houches summer school on dark matter that took place in 2021. The main topic is dark matter direct detection, and there are two main lectures. In the first one, the author presents a first-principle calculation for the scattering cross section within a specific model where dark matter interactions are mediated by a spin-one particle. In the second lecture, the author develops an effective theory where dark matter elastic scatterings are mediated by contact interactions. Particular relevance is given to the scale separation and the connection via renormalization group evolution. Both dark matter particles and target nuclei are non-relativistic in the process, and the appropriate framework for both is developed along the lines of heavy-quark effective theory and chiral perturbation theory, respectively.

I find these notes well written. The detailed calculation in section 2 is something I had never seen in the literature and that I had to derive myself the first time I did it. I believe this is very useful. Through the lecture notes, few exercises are suggested to the reader, and solutions can be found in the appendix. I am happy to recommend this paper for publication.

---

## Round 1 · Referee Report · Anonymous · 2021-10-5

Report

This is a nice set of lecture notes on an important application of effective field theory. The notes do a very nice job showing how EFT is a significant simplification in this case, by going through the standard calculation of some of the results in detail before going through them again with EFT techniques.

The chiral effective theory subsection was pretty spare and I wonder if it would really be sufficient for anyone seeing it for the first time to know what is going on. But it is also not necessary for most of the rest of the lectures, and moreover the lectures by Pich and Georgi on chiral EFT are mentioned in the introduction (perhaps these two references should be mentioned again on page 10).

I thought the comment about the pion propagator below equation (68) should be explained a bit more carefully. The pion propagator more precisely is 1/(q^2+m_pi^2), and in DM direct detection processes it is common for q to be similar to or smaller than m_pi, so saying the propagator is 1/q^2 could cause confusion. The author explicitly calculates the coefficients of O1 and O6 in equations (74),(75) and I think referring to these would also help clarify this point. One minor comment is that I did not see it explicitly stated anywhere that the (lower-case) c_j's are the coefficients of the operators O_j in (68) in the effective Lagrangian.

Overall, these are a well-written and useful introduction to the material.

  • validity: top
  • significance: high
  • originality: good
  • clarity: high
  • formatting: excellent
  • grammar: excellent

Author:  Joachim Brod  on 2021-11-19  [id 1959]

(in reply to Report 2 on 2021-10-05)

Dear referee, thank you for your helpful comments. All your suggestions will be incorporated in an updated version of the lecture notes.

---

## Round 1 · Referee Report · Anonymous · 2021-11-5

Report

The manuscript "Dark Matter Effective Theory" by Joachim Brod, summaries the content of a short course taught at Les Houches 2021. The course consists of two lectures. The first one describes how to calculate the dark matter-nucleus scattering cross section within a specific model for dark matter-quark interactions. The second lecture generalises and extends the content of the first lecture by applying heavy field expansions and chiral perturbation theory to the problem of dark matter scattering by nuclei. Both lectures are complemented by a set of exercises solved in an appendix.

The material in the course is well structured and presented in a consistent way. The first lecture is very clear, and many students will benefit from it. The second one is more technical, and the students will have to complement it with further literature studies.

Based on these considerations, I can recommend this work for publication. I only have some minor comments and suggestions which I list below:

- The estimate of the transferred energy below Eq. (6) doesn't seem to be correct. With the given parameter values, it should be 200 keV. Indeed, the quoted value, 2 keV, is close the energy threshold of operating direct detection experiments.

- Why (q'-q) in the exponential in the second line of Eq. (20)? Shouldn't be (k_2-k_1)?

- The statement "Since the Lagrangian must be a dimensionless number" is misleading. If by Lagrangian the author means Lagrangian density, as in the text above Eq. (31), then a Lagrangian density must have dimension mass to the fourth.

- It could be worth expanding the part on the spurion method, as it plays a key role in the discussion.

- Add a references for Eq. (52).

- The coupling constants c_N appearing in Eq. (69) should be defined in the text. How to obtain their relation to the relativistic coupling constants, Eqs. (73) and (75), could also be explained more in detail.

- The word "here" appears two times in the sentence "here I list some introductory material (lectures, reviews, and original articles) to the various topics here". The author might consider reformulating this sentence.

- Typo in the sentence "Instead of writing the down the full basis".

  • validity: -
  • significance: -
  • originality: -
  • clarity: -
  • formatting: -
  • grammar: -

Author:  Joachim Brod  on 2021-11-19  [id 1960]

(in reply to Report 3 on 2021-11-05)

Dear referee, thank you for your many detailed comments and pointing out errors and typos. They have all been incorporated / corrected in an updated version of the lecture notes. In particular, I have been more careful in distinguishing Lagrangian and Lagrangian density (where it is relevant), and expanded the discussion of the spurion method for the quark mass terms. I decided not to explicitly include the lengthy detailed discussion of the spurion method for the DM interactions, and kept the reference to the literature instead. I hope, however, that the spirit of the method is more clear in the updated version.

---

## Editorial Decision

published